# Genome-Wide Identification of Strawberry C2H2-ZFP C1-2i Subclass and the Potential Function of *FaZAT10* in Abiotic Stress

**DOI:** 10.3390/ijms232113079

**Published:** 2022-10-28

**Authors:** Hao Li, Maolan Yue, Leiyu Jiang, Yongqiang Liu, Nating Zhang, Xiaoling Liu, Yuyun Ye, Ximeng Lin, Yunting Zhang, Yuanxiu Lin, Mengyao Li, Yan Wang, Yong Zhang, Ya Luo, Xiaorong Wang, Qing Chen, Haoru Tang

**Affiliations:** 1College of Horticulture, Sichuan Agricultural University, Chengdu 611130, China; 2Chengdu Academy of Agriculture and Forestry Sciences, Chengdu 611130, China; 3Institute of Pomology and Olericulture, Sichuan Agricultural University, Chengdu 611130, China

**Keywords:** strawberry, C2H2-ZFPs, abiotic stress, ZAT10

## Abstract

C2H2-type zinc finger proteins (C2H2-ZFPs) play a key role in various plant biological processes and responses to environmental stresses. In *Arabidopsis*
*thaliana*, C2H2-ZFP members with two zinc finger domains have been well-characterized in response to abiotic stresses. To date, the functions of these genes in strawberries are still uncharacterized. Here, 126 C2H2-ZFPs in cultivated strawberry were firstly identified using the recently sequenced *Fragaria* *× ananassa* genome. Among these C2H2-ZFPs, 46 members containing two zinc finger domains in cultivated strawberry were further identified as the C1-2i subclass. These genes were unevenly distributed on 21 chromosomes and classified into five groups according to the phylogenetic relationship, with similar physicochemical properties and motif compositions in the same group. Analyses of conserved domains and gene structures indicated the evolutionary conservation of the C1-2i subclass. A Ka/Ks analysis indicated that the C1-2i members were subjected to purifying selection during evolution. Furthermore, *FaZAT10*, a typical C2H2-ZFP, was isolated. *FaZAT10* was expressed the highest in roots, and it was induced by drought, salt, low-temperature, ABA, and MeJA treatments. It was localized in the nucleus and showed no transactivation activity in yeast cells. Overall, these results provide useful information for enriching the analysis of the ZFPs gene family in strawberry, and they provide support for revealing the mechanism of *FaZAT10* in the regulatory network of abiotic stress.

## 1. Introduction

Various abiotic and biotic stresses affect plant physiology and growth [1]. Severe drought occurred in the northern hemisphere in 2022, and the Yangtze River basin experienced the longest drought since records began in the 1960s. Strawberry (*Fragaria × ananassa*) is widely cultivated all over the world. It plays an important role due to its richness in minerals, vitamins, flavonoids, and other nutrients [2]. Additionally, it is desirable due to its bright color, enjoyable taste, short growth cycle, and high economic benefits [3]. However, it is easily affected by unfavorable environments, such as drought, high salinity, and extreme temperature, during production. These abiotic stresses eventually lead to reductions in strawberry yield and quality, and they are limiting factors for strawberry production [4,5].

To combat the detrimental effects of stressors, plants have evolved effective defense mechanisms by influencing their tolerance potential through integrating molecular- and cellular-level responses [6]. Typically, abiotic stress induces the production of excess reactive oxygen species (ROS) and causes oxidative damage to plants. After receiving ROS signals, plants activate both enzymatic and non-enzymatic scavengers, such as SODs, CATs, APXs, and anthocyanins, to scavenge excess ROS [7,8,9]. In addition, transcription factors (TFs), such as MYB, bHLH, WRKY, bZIP, NAC, AP2/ERF, and ZFP, are also key components of the plant stress response mechanism. Among them, the TFs that contain a zinc finger domain are widely involved in abiotic stress responses [10,11,12,13,14,15,16].

Zinc finger proteins (ZFPs) are one of the largest families in plants, and they play essential roles in many cellular functions, including transcriptional activation and inhibition, RNA binding, apoptosis regulation, and protein interaction [17]. ZFP TFs are divided into nine subfamilies, namely, C2H2, C2HC, C2HC5, C3H, C3HC4, C4HC3, Cys4, C6, and C8, according to the sequence and number of cysteine (Cys) and histidine (His) residues in the ZFP structure [17,18]. In *Arabidopsis*
*thaliana*, there are 176 C2H2 zinc finger proteins (C2H2-ZFPs) divided into three groups (A, B, and C), and the C group is further classified into C1, C2, and C3 subsets. The C1 subset is one of the evolutionarily youngest families, comprising 64 members whose biological functions are related to developmental processes and stress responses [19]. The C1 subset can be subdivided into five subclasses, namely, C1-1i, C1-2i, C1-3i, C1-4i, and C1-5i (Ni indicates N zinc fingers), with 33, 20, 8, 2, and 1 members, respectively [18,19]. Among these subclasses, the C1-2i subclass has been the most extensively studied, and it has been demonstrated to be involved in plant development and stress responses; for example, *AZF1/2/3*, *AtZAT6*, *AtZAT18*, and *AtZAT12* of *A. thaliana* have been found to be involved in water, salt, drought, and strong light stresses [20,21,22,23].

*ZAT10*, formerly known as *STZ* (salt tolerance zinc finger), belongs to a typical C2H2-ZFP, consisting of two C2H2-type zinc fingers CX_2-4_CX_3_FX_5_LX_2_HX_3-5_H (X: any amino acid; number: number of amino acids), and it contains two conserved Cys and His residues and an EAR (L/FDLNL/F(x)P) motif at its C-terminus [24,25,26]. Previous studies have reported that the constitutive expression of *ZAT10* enhances tolerance to salt, heat, and osmotic stresses in *A. thaliana*. Interestingly, the knockout and RNAi mutants of *AtZAT10* have also been found to be more tolerant to osmotic and salt stresses [24]. It has been demonstrated that the EAR motif of *ZAT10* can function as a transcriptional repressor [27,28,29]; inhibiting the expression level of *ZAT10* by RNAi or knockout mutation will lead to enhanced tolerance if this motif is directly involved in the suppression of stress [24]. These results suggest that *ZAT10* may have different regulatory mechanisms in response to abiotic stress in *A. thaliana*. 

At present, most of the studies on C2H2-ZFPs in abiotic stress are still focused on model plants, such as *A. thaliana*. In cultivated strawberries, the family members of C2H2-ZFPs and their regulatory mechanisms in response to abiotic stress are largely unknown. In the present study, C2H2-ZFPs in cultivated and wild strawberries were identified based on genomic data. Among these C2H2-ZFPs, the phylogenetic relationship, protein physicochemical properties, gene structure, conserved domains, promoter cis-acting elements, and gene collinearity of the C1-2i subclass were further analyzed. Then, *FaZAT10* from the C1-2i subclass was isolated. The expression pattern, subcellular localization, and transcriptional activity of *FaZAT10*, as well as its expression pattern under different abiotic stresses and hormone treatments, were further analyzed. These results improve the characterization of strawberry C2H2-ZFP family members and provide a research basis for the further exploration of how *FaZAT10* plays a role in strawberry abiotic regulatory networks.

## 2. Results

### 2.1. Identification and Chromosomal Localization of Strawberry C2H2-ZFP C1-2i Subclass

A total of 126 and 41 C2H2-ZFP candidate genes were identified in cultivated and wild strawberries, respectively. Among these, the following subclasses were identified: the 5i subclass (3 FaZATs and 1 FvZAT); the 4i subclass (11 FaZATs and 2 FvZATs); the 3i subclass (29 FaZATs and 8 FvZATs); the 2i subclass (46 FaZATs and 13 FvZATs); and the 1i subclass (37 FaZATs and 17 FvZATs). In the following work, the C1-2i subclass in cultivated strawberry was further studied. The nomenclature of the C1-2i subclass in cultivated strawberry was based on the homologous genes of diploid wild strawberry in NCBI, and the same alleles were numbered according to their position on chromosomes 1-7 (Figure 1 and Appendix A).

The C1-2i subclass in cultivated strawberry was unevenly distributed on 21 chromosomes. The density of the genes was the highest on chromosomes Fvb3-2, Fvb3-3, and Fvb6-1, with four genes each. Only one gene was found on Fvb2-4, Fvb3-1, Fvb4-1, Fvb5-1, Fvb5-2, Fvb5-3, and Fvb5-4. Most of the *FaZAT* genes were in the regions at both ends of the chromosome (Figure 1).

### 2.2. Characterization of the C2H2-ZFP C1-2i Subclass

In the C1-2i subclass, the proteins ranged from 158 to 383 amino acids in length. The theoretical isoelectric point (pI) was between 6.39 and 10.14. The pIs of most members were greater than seven, indicating that they contained more basic amino acids. In addition, the molecular weights of all C1-2i proteins ranged from 17,378 to 42,826 Da. The predicted results of subcellular localization indicated that they were all located in the nucleus. The GRAVY values ranged from -0.895 to -0.198, revealing that strawberry C1-2i proteins were hydrophilic proteins. The instability index ranged from 39.08 to 75.27 (Table 1).

### 2.3. Phylogenetic Relationship and Sequence Alignment of the C2H2-ZFP C1-2i Subclass

To further analyze the evolutionary relationship of the C1-2i subclass in cultivated strawberry, CLUSTALW was used to perform a multiple sequence alignment of 46 genes of the 2i subclass in cultivated strawberry and 13 in wild strawberry. Then, 20 C1-2i genes from *A. thaliana* [19] were introduced to construct a phylogenetic tree, as shown in Figure 2. The 46 C1-2i subclass members in cultivated strawberry and the 13 C1-2i subclass members in wild strawberry were mainly distributed in five groups: Group I (12 FaZATs and 5 FvZATs), Group II (7 FaZATs and 2 FvZATs), Group III (15 FaZATs and 3 FvZATs), Group IV (7 FaZATs and 2 FvZATs), and Group V (5 FaZATs and 1 FvZAT). The results of the multiple sequence alignment showed that most members contained a conserved motif, QALGGH, in the zinc finger helix (Appendix A), which is essential for the DNA-binding activity of C2H2-ZFPs [16].

### 2.4. Gene Structure and Conserved Domain Analysis

The gene structure of the C1-2i subclass changed irregularly; most genes contained two introns and one exon. In particular, *FaZAT8b* and *FaZAT* contained a 10,622 bp-long and 7559 bp-long open reading frame (ORF), respectively (Figure 3B and Table 1). Furthermore, three highly conserved motifs were analyzed on the MEME online website, including 29 conserved amino acid residues in Motif 1, 41 in Motif 2, and 21 in Motif 3. A sequence analysis showed that Motif 1 and Motif 2 were C2H2 domains and that Motif 3 was an EAR motif at the C-terminal, which is consistent with C1-2i member characteristics, indicating that these genes were evolutionarily conserved (Appendix A). 

### 2.5. Promoter Analysis of the C2H2-ZFP C1-2i Subclass Members in Strawberry

The cis-acting regulatory elements in the 2000 bp promoter region were predicted using the PlantCARE website. The results show that a total of 62 types of cis-regulatory elements were identified, except for the core promoter elements of a higher plant TATA-box and the universal promoter enhancement element CAAT-box, as well as unannotated elements. Among them, light responses were the most abundant, with 438 existing, followed by 182 MeJA-responsive elements and 124 ABA-responsive elements. Meanwhile, 15 elements related to plant development and 6 related to stress response were analyzed (Figure 4 and Appendix A). In addition, MYB-related and MYC-related elements were also found to be abundant in the promoter region of the C1-2i subclass members in cultivated strawberry.

### 2.6. Synteny Analysis of the C2H2-ZFP C1-2i Subclass in A. thaliana, F. vesca, and F. × ananassa

There were 57 syntenic gene blocks of the C1-2i genes among the cultivated strawberry chromosomes, and 61 *FaZAT* gene pairs were confirmed as collinear pairs (Appendix A). To further explore the evolution of the C1-2i gene, interspecific synteny comparisons between *F. × ananassa*, *F. vesca*, and *A. thaliana* were also made. There were 91,807 and 42,389 collinear genes observed between *F. × ananassa* and *F. vesca* and between *F. × ananassa* and *A. thaliana*, respectively, with ratios of 64.45% and 31.17%. The interspecies collinearity analysis indicated that there may have been chromosomal and gene amplification events during the evolution of the two strawberries (Figure 5 and Figure 6). 

The ratio of nonsynonymous substitution rates (denoted as Ka) to synonymous substitution rates (denoted as Ks) was used to assess the selective pressure on duplicate gene pairs within species. Ka/Ks >1 indicated positive selection, and Ka/Ks <1 indicated purifying selection [30]. The driving force of C1-2i gene pairs on replication was calculated using the Ka/Ks ratio. The Ka/Ks values of the C1-2i gene pairs were less than one, suggesting that these genes were influenced by purifying selection during evolution (Appendix A).

### 2.7. Transcriptome Analysis of Strawberry C2H2-ZFP C1-2i Subclass under Salt Stress

To better understand the response of *FaZATs* to stress at the transcriptional level, transcriptome data were used to detect the transcript abundance of *FaZATs* under salt stress. The results show that, under salt stress, *FaZAT10*, *10a*, *10b*, *10c*, *10d*, *10e*, *10f*, *FaZAT12*, *12a*, *12c*, *12d*, *FaZAT8*, *8a*, *8b*, *FaZAT11a*, *11b*,*11c*, *11f*, and *11i* were highly expressed in roots compared to the control (Figure 7A). In particular, the transcriptome data showed that *FaZAT10* exhibited the highest expression in roots under salt stress (Figure 7B and Appendix A).

### 2.8. Tissue-Specific Expression Profile of FaZAT10

Based on the transcriptome data, *FaZAT10* was cloned and studied in the following work. The results of the qRT-PCR analysis suggested that *FaZAT10* was highly expressed in roots, followed by leaves and stems, and the expression level was the lowest in runners. In general, the transcript abundance of *FaZAT10* was very low in different fruit developmental stages, and it had a higher expression level at the little green stage than at the other fruit developmental stages (Figure 8).

### 2.9. The Subcellular Localization and Transcriptional Activity Analysis of FaZAT10

The GFP fluorescence signal of the FaZAT10::GFP fusion protein was located in the nucleus, which indicates that FaZAT10 is a nucleus-localized protein (Figure 9A). In addition, the result indicates that the yeast strains that transformed with pGBKT7-FaZAT10 (validation group) and pGBKT7-Lam (a negative control) could not grow normally on SD/-Trp/-His/-Ade/X-α-gal and SD/-Trp/-His/-Ade media. The positive group pGBKT7-FaBBX22 [31] grew well on SD/-Trp/-His/-Ade/X-α-gal medium and turned blue, which indicates that *FaZAT10* has no trans-acting ability in yeast cells (Figure 9B).

### 2.10. FaZAT10 Was Induced by Abiotic Stress

Several cis-elements related to abiotic stress were found on the cloned 1,952 bp of the *FaZAT10* promoter sequence, such as the DRE core (stress-induced), ABRE (in response to abscisic acid), LTR (in response to low temperature), MBS (in response to drought), CGTCA motif, and TGACG motif (in response to methyl jasmonate). We hypothesized that *FaZAT10* would respond to these abiotic stresses, and, thereupon, abiotic stress and hormone treatments were designed to detect the expression of *FaZAT10* (Appendix A and Table 2).

The result indicates that, under simulated drought stress, the expression of *FaZAT10* increased continuously in roots and petioles, and it reached the highest level at 24 h. In leaves, the peak increased at 12 h and then gradually decreased. Under salt stress, the expression of *FaZAT1*0 in the different tissues gradually increased along with the treatment time, and it reached the peak at 24 h, especially in roots. The expression pattern of *FaZAT10* was quite different under low-temperature stress. It was significantly elevated in leaves after 3 h of stress, and then it decreased. The growth trend was similar in roots and petioles, and both reached the maximum at 24 h. Under the treatments of ABA and MeJA, *FaZAT10* had no obvious tendency in roots, but the expressions of *F**aZAT10* in petioles and leaves both reached their peaks after 9 h of treatments, and then they decreased. Overall, these results suggest that *FaZAT10* is more sensitive to salt, low-temperature, and drought stress treatments than to ABA and MeJA treatments.

## 3. Discussion

Transcription factors (TFs) play a central role in plant abiotic stress response networks. Currently, based on genome-wide analyses, different TF families have been identified in different plants. These TFs perform functions in various biological processes of plants, and some studies have shown that they are involved in plant abiotic stress responses [10,11,12,13,14,15,32,33]. C2H2-ZFPs are one of the best-studied TFs in plant abiotic stress responses, and 189, 109, and 321 C2H2-ZFPs have been identified in rice (*Oryza sativa*), poplar (*Populus trichocarpa*), and soybean *(Glycine max*), respectively [33,34,35,36,37]. Cultivated strawberry has a high economic value and is vulnerable to the negative effects of abiotic stress. Although *A. thaliana* provides an ideal model for studying abiotic stresses, no C2H2-ZFP family members have been reported in cultivated or wild strawberries to date. Accordingly, the whole-genome sequencing of the octoploid strawberry completed in 2019 and the re-annotation of the octoploid strawberry genome published in 2021 provided us with effective tools for a genome-wide analysis of the C2H2-ZFPs [38,39]. In this study, 126 and 41 C2H2-ZFPs in cultivated and wild strawberries were identified, and they contained one to five scattered zinc finger structures, with a maximum of five C2H2 zinc finger domains and a minimum of one. This identification is consistent with the zinc finger characteristics of subclass C1 reported in *A. thaliana* [17,19]. 

Knowing phylogenetic relationships among species is fundamental for many studies in biology. An accurate phylogenetic tree helps us to infer the origin of new genes and to understand morphological character evolution [40]. Our results show that genes in the same group are more similar to each other. They had similar but not identical protein physicochemical properties, conserved domains, and motifs, suggesting that they may have similar biological functions. For example, *AtZAT6*, *AtZAT10*, and *AtZAT13* in the fourth group may be involved in the response to drought, salt, and low-temperature stress [19]. Moreover, in the analysis of the domains and motifs, we found that *FaZAT4*, *4a*, *4b*, and *4c* were had characteristics consistent with the C2H2 domain, but their first zinc finger domain did not contain the conserved motif QALGGH and instead contained RALGGH (Appendix A). From the perspective of the zinc finger structure, the arbitrary amino acid X in the CX_2-4_CX_3_FX_5_LX_2_HX_3-5_H domain was significantly different. The presence of a highly conserved motif, QALLGGH, in the zinc finger helix is unique to plants, and the variable spacing between adjacent zinc fingers and each amino acid in the conserved sequence is thought to be important for DNA-binding activity, suggesting that this class of zinc finger proteins is involved in unique plant life processes [17,41,42]. Zinc finger helices containing this conserved motif are referred to as Q-type ZFPs, and those without any conserved motif are called C-type ZFPs [35]. This feature may affect the DNA-binding activity of these four members, which needs to be further demonstrated in experiments. We also found that the last amino acid in the first zinc finger structure of *FaZAT11* is Q instead of H (Appendix A), but it conforms to other characteristics of zinc fingers, so *FaZAT11* is still considered to be a C2H2-ZFP.

Plant cells have subcellular compartments, such as the nucleus, mitochondria, and chloroplast. The localization of plant proteins within cellular compartments was found to be essential in revealing their functions [43,44]. The prediction of the subcellular localization of the C1-2i members in cultivated strawberry showed that they were all localized to the nucleus. At the same time, FaZAT10 is also located in the nucleus. Previous studies in *Arabidopsis* have demonstrated that *AtZAT10* functions as a transcriptional repressor due to the presence of the EAR domain at the C-terminus [28,29]. The EAR motif reduces the underlying transcriptional level of the reporter gene, and the transcriptional activation activity of other TFs [45]. The C-terminal inhibitory residues of *AtERF4* are DLDLNL, and if a mutation occurs in this domain, the protein’s inhibitory function decreases or disappears [46]. The full-length FaZAT10 protein has no transactivation ability in yeast cells, and we conjectured that the EAR motif may be the reason why FaZAT10 does not have transactivation ability, as it may function as a transcriptional repressor in the abiotic stress signal transduction pathway. In cultivated strawberry, whether the DLNL sequence in the EAR motif of *FaZAT10* is a necessary sequence for inhibitory activity needs further study.

The analysis of promoter regions is helpful to study the transcriptional regulation of TFs. The identification of plant promoters often involves the identification and characterization of genes expressed under specific tissue or physiological stress conditions [47]. Abiotic stress treatments were performed according to the cloned *FaZAT10* promoter sequence. Our results indicate that *FaZAT10* was continuously induced in roots and petioles; most pronounced in roots under salt and drought stresses; significantly induced in leaves after 3 h of stress at 4 °C; and significantly induced in petioles after 9 h of MeJA and ABA treatments (Figure 10). In *Arabidopsis*, the expression of *AtZAT10* in leaves was induced by low temperatures, UV-B, oxidative stress, osmotic stress, and genotoxic stress, and it was strongly induced by low-temperature and salt stresses in roots [24]. It was also induced by a high amount of light, ABA, gibberellin, SA, and MeJA [47]. In apple, 10% PEG 6000, 150 mM NaCl, wounding, and 100 μM SA treatment induced the expression of *MdZAT10* [48]. Our results are basically consistent with previous studies. The transcriptome data showed that only *FaZAT10* was strongly expressed in the leaves and roots of all C1-2i members under salt stress, suggesting that *FaZAT10* is more likely to play a key role in stress signal transduction. In *Arabidopsis*, Los2 binds to the promoter of *ZAT10* in order to inhibit the expression of *ZAT10*, thereby attenuating the inhibitory effect of *ZAT10* on *RD29A* and positively regulating the expression of *RD29A*, thus achieving the optimal response to cold environments [27]. The overexpression of *AtZAT10* increased tolerance to salt and dehydration stresses and resulted in growth retardation in transgenic *Arabidopsis thaliana* [28]. In apple, the overexpression of *MdZAT10* negatively regulates drought tolerance by regulating the expression of *MdAPX2*, increasing its sensitivity to PEG6000 treatment, and decreasing the ability to scavenge ROS [49]. In our study, *FaZAT10* was more sensitive to salt, low-temperature, and drought stresses than hormone treatments, which indicates its potential function. However, the regulatory mechanism of *FaZAT10* in response to abiotic stress remains to be further investigated.

## 4. Materials and Methods

### 4.1. Identification and Phylogenetic Analysis of 2i Subclass in Strawberry 

The gene sequences of the Arabidopsis C2H2 family were downloaded from TAIR (https://www.arabidopsis.org/) (accessed on 13 July 2022), and the Hidden Markov Model (HMM) profile of the conserved C2H2 domain (PF13912) was obtained from the Pfam database (http://pfam.xfam.org/) (accessed on 13 July 2022). To identify the ZAT genes in strawberries, wild (Fragaria_vesca_v4.0.a2) and cultivated (Fragaria x ananassa Camarosa Genome v1.0.a1) strawberry genome data were downloaded from GDR (https://www.rosaceae.org/) (accessed on 13 July 2022). Then, Simple HMM Search in Tbtools (Version 1.098752) [50] was used to pick out the candidate genes, and the redundant sequences were manually removed. Finally, all candidate genes were examined and analyzed using the SMART (http://smart.embl.de/#) (accessed on 13 July 2022) conserved domain prediction software Pfam (Pfam: Search Pfam (xfam.org)) (accessed on 13 July 2022), and NCBI CDD (https://www.ncbi.nlm.nih.gov/Structure/bwrpsb/bwrpsb.cgi) (accessed on 13 July 2022) was used to ensure that they all contained the conserved C2H2 domain.

The ZAT proteins of *Arabidopsis thaliana*, the cultivated strawberries, and the wild strawberries were aligned using CLUSTALW, and a phylogenetic tree was developed on MEGA (version 7.0) using the neighborhood-joining phylogenetic method analysis. Both the bootstrap test and the approximate likelihood ratio test were set to 1000 times. The nomenclature of the ZAT gene in cultivated strawberry refers to the nomenclature of wild strawberry in NCBI. The same alleles were sequentially numbered according to their position on the chromosome, and the phylogenetic tree was embellished with iTOL (https://itol.embl.de/) (accessed on 15 July 2022).

### 4.2. Characteristic Analysis of the C1-2i Subclass

Online tools (https://web.expasy.org/protparam/) (accessed on 22 July 2022) were used to analyze the protein physicochemical properties of the cultivated strawberry 2i members: Plant-mPLoc (www.csbio.sjtu.edu.cn/cgi-bin/EukmPLoc2.cgi) (accessed on 22 July 2022) was used for subcellular localization prediction; a domain analysis was performed on the NCBI Conserved Domain Database (CDD: https://www.ncbi.nlm.nih.gov/Structure/bwrpsb/bwrpsb.cgi) (accessed on 24 July 2022); and the motifs of cultivated strawberries were analyzed using the MEME Suite (https://meme-suite.org/meme/) (accessed on 24 July 2022). The physical gene location of the ZAT protein in cultivated strawberry was extracted from the cultivated strawberry genome annotation file.

The promoter sequence 2000 bp upstream of the ZAT start codon was extracted with Tbtools. The prediction of cis-acting elements on promoters was identified using the PlantCARE (http://bioinformatics.psb.ugent.be/webtools/plantcare/html/) (accessed on 26 July 2022) online website.

The nonsynonymous mutation rate (denoted as Ka), synonymous mutation rate (denoted as Ks), and the ratio of the nonsynonymous mutation rate to synonymous mutation rate (denoted as Ka/Ks) were calculated using Tbtools. Promoter, gene structure, conserved domains, chromosomal location, and collinearity analysis were all analyzed and visualized using Tbtools [50].

### 4.3. Plant Material and q-PCR Analysis

Octoploid cultivated strawberry ‘Benihoppe’ (*Fragaria × ananassa* Duch.) was asexually propagated using runners. The explant surface sterilization procedure was as follows: 70% alcohol soaking for 30 s, pouring out the alcohol, rinsing with sterile water two times, then soaking with 0.1% HgCl_2_ for 30 min, and rinsing with sterile water five times. Finally, the sterilized runners were inoculated into MS medium (0.5 mg/L 6-BA, 0.1 mg/L IBA), cultured for 45 days, and then transplanted to 1/2 MS medium. Plant material was cultured in a plant material culture room and a greenhouse under long-day conditions (16 h light, 5000 lx, 8 h dark, 22 °C). Plant material was grown in 1/2 MS medium before transplanting, and uniformly growing plants were transplanted in plastic pots with peat soil, perlite, and vermiculite (volume ratio = 4:1:1) and fertilized with regular irrigation.

Different tissues (roots, stems, functional leaves, flowers, ripe red fruits, and runners) of ‘Benihoppe’ were collected from the greenhouse for tissue-specific expression profiling; fruits at different developmental stages were collected for an expression profile analysis: small green (LG: little green, seven days after fruiting), big green (BG: big green, 15 days after fruiting), WF: white fruit, red (TR: turn red, 1/4 red), half red (HR: half red, 1/2 red), and ripe full red (FR: Full Red). Three biological replicates were established for all tissues, and all replicates were obtained from plants cultured under the same conditions. Each biological replicate contained at least six samples, with three or four samples taken from each plant. The total RNA of all samples was extracted using the modified CTAB method [51]; its integrity was checked using 1% agarose gel electrophoresis, and the concentration and quality were detected using a micronucleic acid protein analyzer. Reverse transcription was performed using a Primer Scripr™ RT Kit (TaKaRa, Dalian, China), with a gDNA removal Primer (TaKaRa, Dalian, China) per 1 µg of total RNA for PCR reactions. On the basis of the SYBR premixed ExTaqTM kit (Takara, Dalian, China), the 10 µL qRT-PCR system comprised 1µL cDNA (5 ng/µL), 0.4 µL forward/reverse primer (10 µM), 5 µL 2 × SYBR mix, and 3.2 µL ddH2O. qRT-PCR was performed on a CFX96 real-time PCR system (Bio-Rad, Hercules, CA, USA). Three technical replicates were used for each sample. The relative expression of genes was calculated using the 2^−∆∆CT^ formula [52], and *FaActin* was used as a calibrated housekeeping gene. The primer sequences are listed in Appendix A. The RNA-seq-based expression levels of the *FaZAT* genes in strawberry were retrieved from the online transcriptomic data (SRA accessions: SRR19165835, SRR19165836, SRR19165837, SRR19165838, SRR19165841, SRR19165842, SRR8298771, SRR8298772).

### 4.4. Gene Clone and Sequence Analysis of FaZAT10

The full-length coding sequence (CDS) of *FaZAT10* was cloned from an octoploid cultured strawberry ‘Benihoppe’, and primers were designed with reference to its homologue (*FvZAT10*, XM_004290912.2). The sequencing results were assembled by the software CLC Genomics Workbench (version 3.6.1) and compared with FvZAT10 using DNAMAN (version 8.0). The genome sequence of F.vesca_v2.0.a1 was used as the reference for promoter cloning, and the sequence of 2000 bp upstream of the initiation codon of *FaZAT10* was cloned with the genomic DNA of cultivated strawberry as the template. The primer sequences are listed in Appendix A.

### 4.5. Transcriptional Activation Activity Analysis of FaZAT10 Protein

The full-length CDS of FaZAT10 was inserted into the yeast expression vector pGBKT7 (BD) using the homologous recombination method. After confirming the positive vector, the transcriptional activity of the FaZAT10 protein was detected by transforming pGBKT7-Lam (a negative control), pGBKT7-FaBBX22 (a positive control) [31], and pGBKT7-FaZAT10 (validation group) into Y2H cold. After three days of culture at 28 °C, the growth of the yeast strains on different defective media (SD/ −Trp and SD/ −Trp-Ade-His /X-α-gal) was observed.

### 4.6. Abiotic Stress and Hormone Treatments

The octoploid cultivated strawberry ‘Benihoppe’ grown in plastic pots was pulled out, and the roots were washed with water and placed in a tissue culture bottle containing Hoagland nutrient solution (Hypo, HB8870-1) for seven days in a balanced culture, during which the nutrient solution was changed once. Low-temperature stress: the balanced octoploid strawberry ‘Benihoppe’ was put into the artificial climate box at 4°C, and the relative humidity was 75% in common Hoagland solution. Drought stress: Hoagland solution +10% PEG6000 simulated drought. Salt stress: Hoagland solution +200 Mm NaCl. Hormone treatment: 250 mM MeJA and 100 μM ABA were sprayed on the foliage with potted seedlings in common Hoagland solution. The control was potted seedlings in common Hoagland solution at 25 °C under long-day conditions. Roots, petioles, and leaves were sampled at 0 h, 3 h, 6 h, 9 h, 12 h, and 24 h. Each treatment contained at least six potted seedlings, with three biological replicates. Afterward, they were flash-frozen in liquid nitrogen at −80 °C for reservation.

### 4.7. Subcellular Localization

The amplified CDS of *FaZAT10* and green fluorescent protein (GFP) was inserted into the plasmid vector (PYTSL-16) by homologous recombination, which was modified with pMDC83-35S and Psite-2NB [53]. The fusion expression of *FaZAT10* and GFP (FaZAT10:GFP) was driven by the 35S promoter, and the nuclear localization maker for HY5-mCherry had the same 35S promoter drive [54]. The plasmid was transformed into Agrobacterium tumefaciens GV3101, and the two vectors were transiently co-expressed in tobacco epidermal cells [55]. All fluorescence signals of the samples were detected using a confocal laser scanning microscope system (FV3000 Olympus, Tokyo, Japan). The primers are listed in Appendix A.

### 4.8. Statistical Analysis

All data were analyzed using IBM SPSS statistical software (version 23). Student’s *t*-test (**, *p* < 0.01; *, *p* < 0.05), one-way analysis of variance (ANOVA), Waller–Duncan test and multiple comparisons (*p* < 0.05) were carried out, and SigmaPlot (version 12.5) was used to plot the graphs. All of the experimental data are expressed as the mean ± standard error of the mean (SE).

## 5. Conclusions

In this study, for the first time, the C2H2-ZFPs in octoploid and diploid strawberry were identified. The C1-2i subclass members that contained two C2H2 domains were further systematically analyzed. These results contribute to our understanding of the evolution of the C1-2i subclass in cultivated strawberry. Meanwhile, *FaZAT10* of the C1-2i subclass was cloned based on stress transcriptome data. Analyses of cis-acting elements in the promoter region and stress induction experiments revealed that *FaZAT10* may play an important role in regulating stress signal transduction. Taken together, these results provide new information on the responses of C2H2-ZFPs to abiotic stress in strawberry. In addition, we preliminarily investigated the transcription factor properties of *FaZAT10*, and the specific molecular mechanism of *FaZAT10* in abiotic stress needs to be further studied.

## Figures and Tables

**Figure 1 ijms-23-13079-f001:**
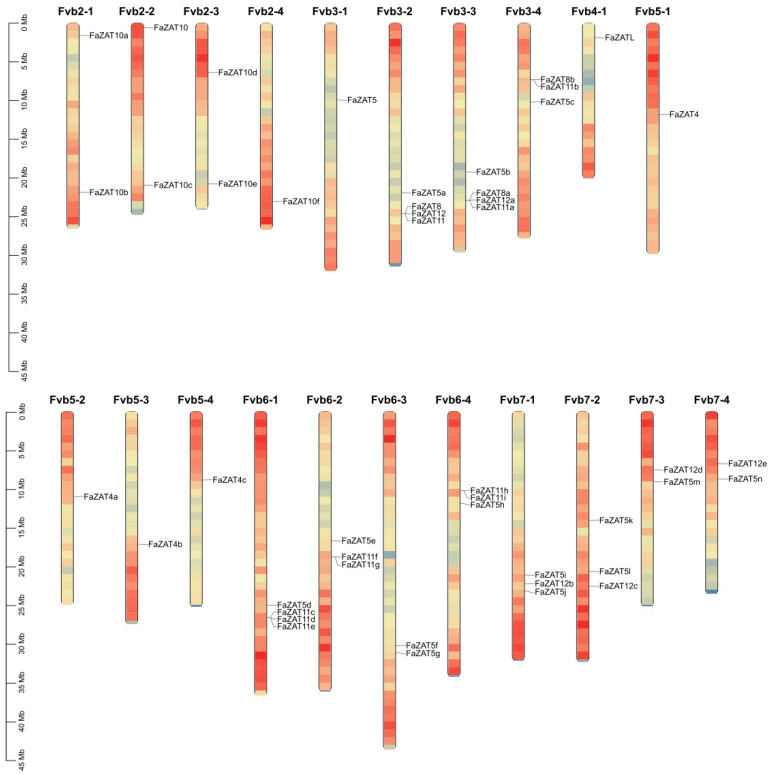
Chromosomal localization of cultivated strawberry C2H2-ZFP C1-2i subclass.

**Figure 2 ijms-23-13079-f002:**
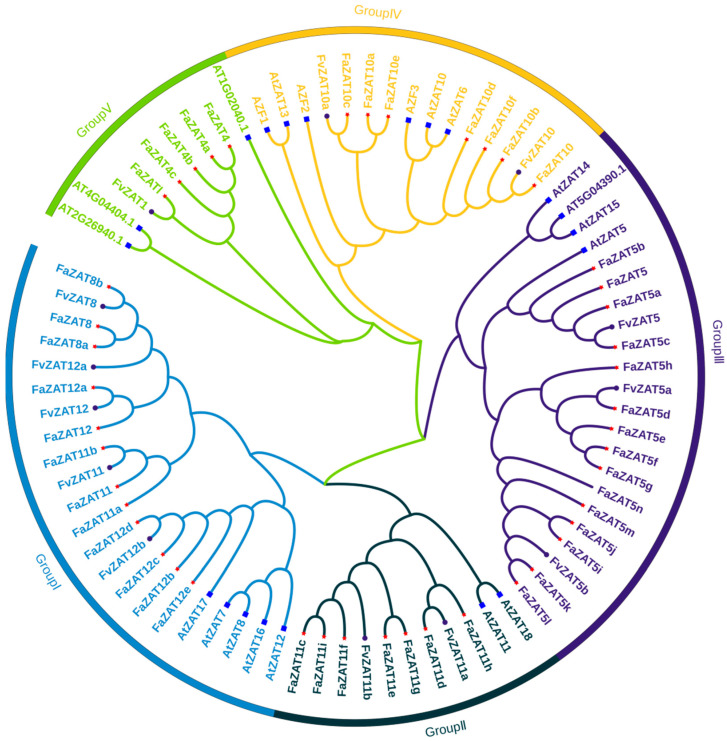
Unrooted phylogenetic tree of cultivated strawberry, wild strawberry, and *A. thaliana* C1-2i subclass members. The red asterisks represent cultivated strawberry, the purple circles represent wild strawberry, and the blue squares represent *A. thaliana*. The phylogenetic tree was developed on MEGA (version 7.0) using the neighborhood-joining phylogenetic method analysis. Both the bootstrap test and the approximate likelihood ratio test were set to 1000 times.

**Figure 3 ijms-23-13079-f003:**
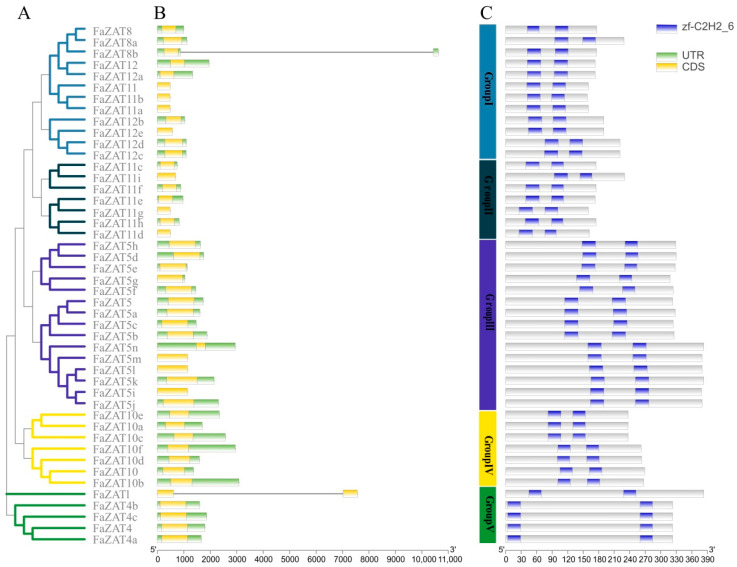
The evolutionary tree (**A**), gene structures (**B**), conserved domains (**C**), and conserved motifs of the cultivated strawberry C1-2i subclass. Blue rectangles are conserved C2H2 domains, yellow rectangles are UTRs, and green rectangles are CDS. The phylogenetic tree was developed on MEGA (version 7.0) using the neighborhood-joining phylogenetic method analysis. Both the bootstrap test and the approximate likelihood ratio test were set to 1000 times.

**Figure 4 ijms-23-13079-f004:**
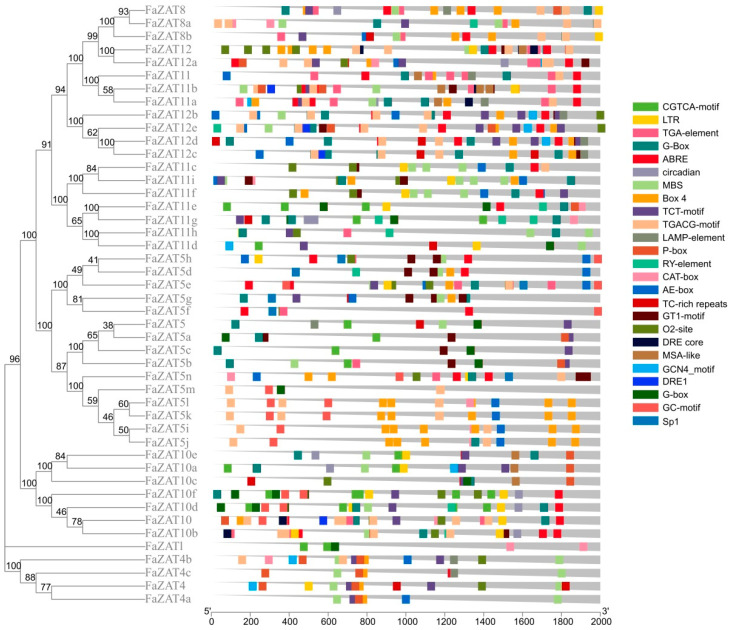
Distribution of cis-acting elements in the promoter of C1-2i subclass members in cultivated strawberry. The phylogenetic tree was developed on MEGA (version 7.0) using the neighborhood-joining phylogenetic method analysis. Both the bootstrap test and the approximate likelihood ratio test were set to 1000 times.

**Figure 5 ijms-23-13079-f005:**
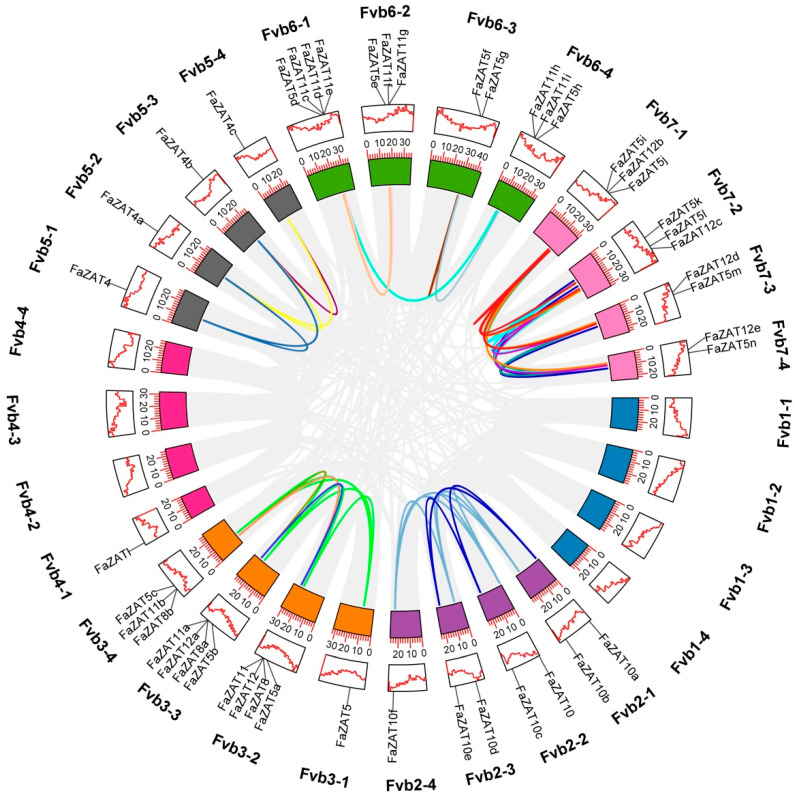
Synteny analysis of C2H2-ZFP C1-2i subclass in cultivated strawberry. Chromosomes are distinguished by different colors, the gray curve is the collinear gene region within the genome, and the colored curve is the collinear gene pair of the C2H2-ZFP C1-2i subclass.

**Figure 6 ijms-23-13079-f006:**
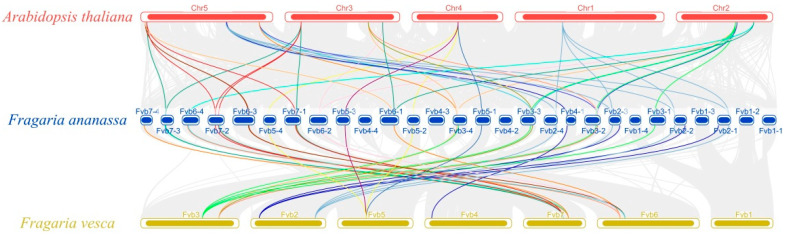
Synteny analysis of C2H2-ZFP C1-2i subclass in *A. thaliana*, *F.* × *ananassa,* and *F. vesca* genomes. The colored rounded rectangles represent the chromosomes of the three plants. The gray curves are the collinear gene regions within the genomes of the three species, and the colored curves represent the gene pairs that are collinear with the C2H2-ZFP C1-2i subclass.

**Figure 7 ijms-23-13079-f007:**
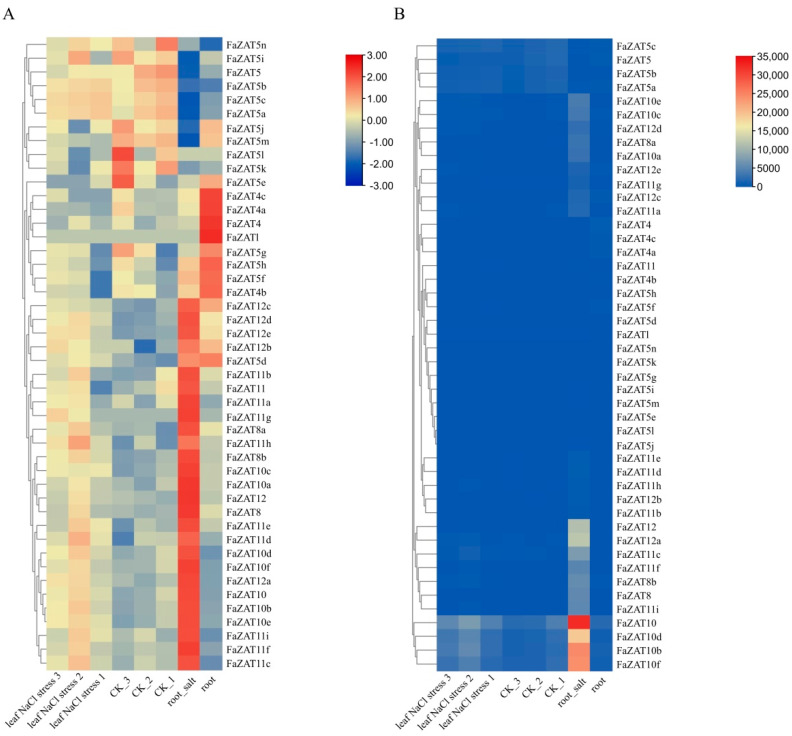
Transcript abundance of the C2H2-ZFP C1-2i subclass under salt stress. NaCl stress: 100 mM NaCl for 12 h; numbers indicate three biological replicates. The heatmap represents log2 value of FPKM. (**A**) The FPKM of each row was normalized. (**B**) The FPKM was not normalized.

**Figure 8 ijms-23-13079-f008:**
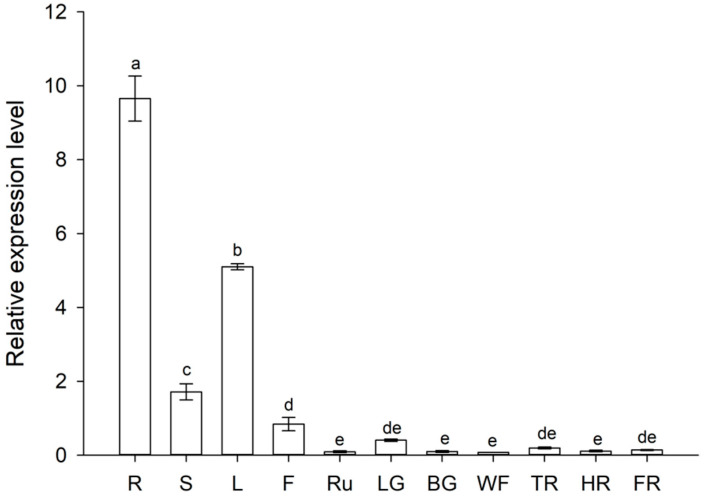
Relative expressions of *FaZAT10* in different tissues and developmental stages of fruits. R: root; S: stem; L: leaf; F: flower; Ru: runner; LG: little green; BG: big green; WF: white fruit; TR: turn red; HR: half red; FR: full red. Data contain three biological replicates, error bars represent standard error of the mean, and different letters are represented by the least significant difference test (LSD), *p* < 0.05.

**Figure 9 ijms-23-13079-f009:**
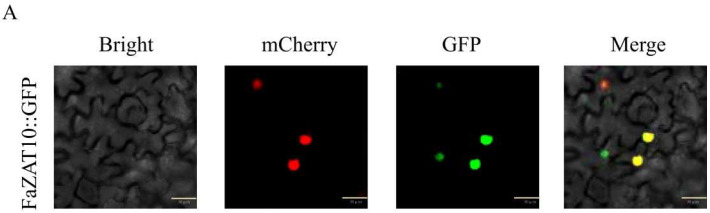
Transcription factor characterization of *FaZAT10*. (**A**) Subcellular localization of FaZAT10 protein. Bars, 30 µm. (**B**) Transcriptional activity analysis of FaZAT10 in yeast cells. pGBKT7-Lam is a negative control, and pGBKT7-FaBBX22 is a positive control.

**Figure 10 ijms-23-13079-f010:**
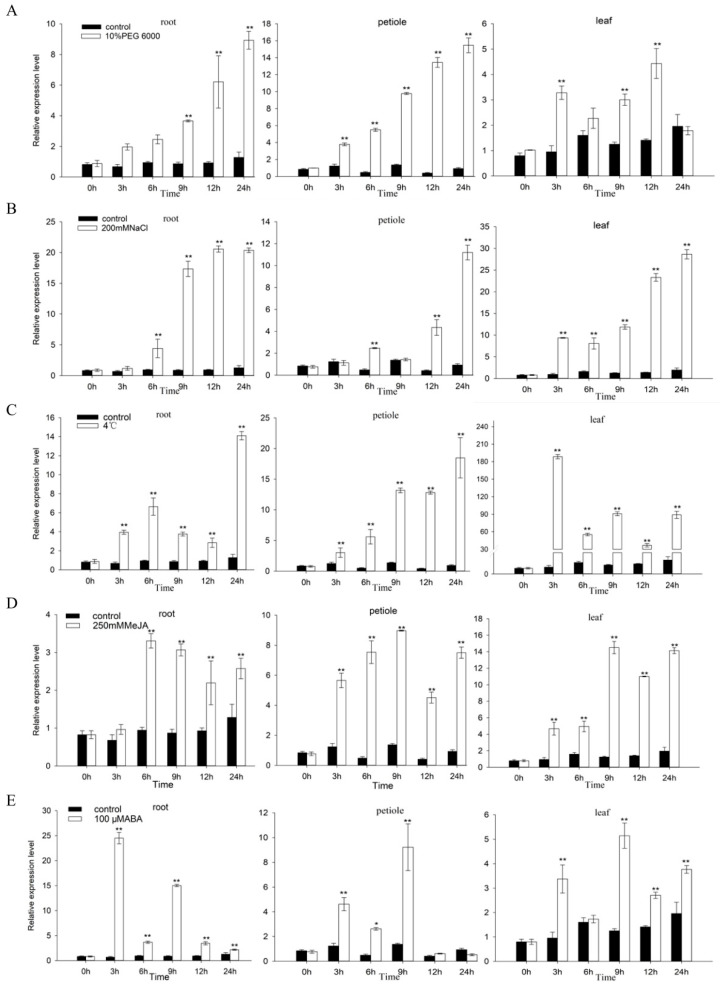
Expression patterns of *FaZAT10* under different abiotic stresses. The relative expression levels of *FaZAT10* under drought (**A**), salt stress (**B**), and low-temperature treatments (**C**) and MeJA (**D**) and ABA (**E**) hormone treatments. Data contain three biological replicates, error bars represent the standard error of the mean, and significant differences between samples were determined using Student’s *t*-test (**, *p* < 0.01; *, *p* < 0.05).

**Table 1 ijms-23-13079-t001:** The physical and chemical properties of C2H2-ZFP C1-2i proteins in *F. × ananassa*.

Gene Name	Amino Acid/aa	ORF/bp	Molecular Weight	pI	Instability Index	GRAVY	Location	Subcellular Localization
FaZAT8b	176	10,622	18,848	9.61	55.77	−0.445	7,282,074–7,292,696	Nucleus
FaZAT8	176	997	18,944	9.60	59.49	−0.475	24,596,211–24,597,208	Nucleus
FaZAT8a	229	1115	24,872	9.70	53.08	−0.264	22,862,614–22,863,729	Nucleus
FaZAT12	173	1955	19,019	9.16	59.78	−0.447	24,604,928–24,606,883	Nucleus
FaZAT12a	173	1300	18,957	9.59	59.83	−0.482	22,874,029–228,75,359	Nucleus
FaZAT11b	158	476	17,378	9.69	67.74	−0.645	7,323,096–7,323,572	Nucleus
FaZAT11	160	482	17,496	9.90	62.24	−0.599	24,610,951–24,611,433	Nucleus
FaZAT11a	160	482	17,586	10.14	60.52	−0.555	22,888,095–22,888,577	Nucleus
FaZAT12d	221	1094	24,382	9.38	62.40	−0.325	7,514,144–7,515,238	Nucleus
FaZAT12c	221	1086	24,400	9.38	63.82	−0.357	22,530,332–22,531,418	Nucleus
FaZAT12b	190	1036	20,593	9.10	60.63	−0.385	22,223,399–22,224,435	Nucleus
FaZAT12e	190	572	20,529	9.17	64.01	−0.452	6,714,293–6,714,865	Nucleus
FaZAT11c	175	752	19,303	9.55	39.08	−0.341	26,593,340–26,594,092	Nucleus
FaZAT11i	230	692	25,364	9.60	40.86	−0.198	10,202,617–10,203,309	Nucleus
FaZAT11f	175	885	19,376	9.44	47.34	−0.430	18,762,037–18,762,922	Nucleus
FaZAT11g	160	482	17,605	9.56	44..84	−0.389	18,771,968–18,772,450	Nucleus
FaZAT11d	162	488	18,229	9.90	42.02	−0.451	26,596,604–26,597,092	Nucleus
FaZAT11e	173	966	19,051	9.51	44.56	−0.378	26,601,748–26,602,714	Nucleus
FaZAT11h	175	824	19,730	9.83	42.85	−0.494	10,199,732–10,200,556	Nucleus
FaZAT5l	380	1142	41,776	6.96	73.60	−0.889	20,647,253–20,648,395	Nucleus
FaZAT5k	383	2148	42,049	6.96	73.11	−0.895	14,054,752–14,056,900	Nucleus
FaZAT5i	379	1139	41,510	6.81	70.02	−0.868	21,096,734–21,097,873	Nucleus
FaZAT5j	380	2310	41,659	6.81	70.30	−0.876	23,197,453–23,199,763	Nucleus
FaZAT5n	383	2944	42,340	6.94	71.63	−0.930	8,687,658–8,690,602	Nucleus
FaZAT5m	380	1142	41,895	6.83	73.37	−0.938	9,080,140–9,081,282	Nucleus
FaZAT5g	318	1041	34,736	6.60	66.74	−0.634	31,122,729–31,123,770	Nucleus
FaZAT5f	324	1449	35,525	6.60	66.78	−0.608	30,224,033–30,225,482	Nucleus
FaZAT5e	328	1126	36,077	6.64	69.83	−0.612	16,681,442–16,682,568	Nucleus
FaZAT5d	330	1750	36,307	6.39	68.39	−0.624	24,991,197–24,992,947	Nucleus
FaZAT5h	329	1626	36,126	6.42	75.27	−0.595	11,865,273–11,866,899	Nucleus
FaZAT5c	324	1465	35,394	7.11	70.08	−0.712	10,177,077–10,178,542	Nucleus
FaZAT5a	328	1609	35,634	7.11	67.31	−0.642	21,919,033–21,920,642	Nucleus
FaZAT5	323	1734	35,198	7.06	65.80	−0.627	9,931,220–9,932,954	Nucleus
FaZAT5b	326	1876	35,598	7.38	66.85	−0.644	19,227,512–19,229,388	Nucleus
FaZAT10	269	1366	28,458	8.50	58.90	−0.565	574,359–575,725	Nucleus
FaZAT10b	267	3084	28,341	8.50	57.18	−0.600	21,865,334–21,868,418	Nucleus
FaZAT10f	262	2951	27,764	8.50	55.29	−0.598	23,032,727–23,035,678	Nucleus
FaZAT10d	263	1590	28,206	8.95	56.46	−0.622	6,379,625–6,381,,215	Nucleus
FaZAT10e	237	2343	25,184	8.45	73.44	−0.557	20,755,846–20,758,189	Nucleus
FaZAT10a	237	1693	25,258	8.65	73.10	−0.516	1,590,173–1,591,866	Nucleus
FaZAT10c	237	2571	25,317	8.15	70.18	−0.578	20,926,689–20,929,260	Nucleus
FaZAT4	323	1797	36,131	6.39	60.22	−0.834	11,788,298–11,790,095	Nucleus
FaZAT4a	323	1655	36,131	6.39	60.22	−0.834	11,007,603–11,009,258	Nucleus
FaZAT4b	323	1595	36,127	6.57	58.62	−0.820	17,138,548–17,140,143	Nucleus
FaZAT4c	323	1862	36,213	6.44	59.36	−0.831	8,825,171–8,827,033	Nucleus
FaZATl	383	7559	42,826	6.46	66.57	−0.786	1,860,056–1,867,615	Nucleus

**Table 2 ijms-23-13079-t002:** Cis-regulatory elements of *FaZAT10* promoter in cultivated strawberry.

Function	Type	Motif
Light-responsive	4	Box4, G-box, G-Box, chs-CMA1a, TCT-motif
Meristem expression	1	CAT-box
ABA response	1	ABRE
Adversity induction	2	DRE core, DRE1
Low-temperature response	1	LTR
Auxin-responsive	1	TGA-element
Gibberellin-responsive	1	P-box
MeJA response	2	CGTCA motif, TGACG motif
Drought stress	1	MBS
Anoxic specific inducibility	1	GC motif
MYB-binding site	2	MYB, Myb-binding site
MYC-binding site	1	MYC
WRKYs response element	1	W-box
Seed-specific regulation	1	RY element
Circadian control	1	Circadian

## Data Availability

Not applicable.

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
