# Peer review of "Genome-Wide Identification of Strawberry C2H2-ZFP C1-2i Subclass and the Potential Function of FaZAT10 in Abiotic Stress"

_ijms, 2022, doi:10.3390/ijms232113079_

Round 1

Reviewer 1 Report

Comments and Suggestions for Authors

  The manuscript entitled "Genome-wide identification of strawberry C2H2-ZFPs C1-2i subclass and reveal the potential function of FaZAT10 inabiotic stress". It is reviewed, this study is comprehensive research. This study is interesting and useful. But there are some deficiencies are in manuscript. By this, minor revision required for article. The paper should be thoroughly revised (typing and grammar mistakes and another mistakes).The manuscript is recommended for publication.

 My minor comments are (Suggestions to improve):

 1-      Please, explain why did you use this material.

2-      Check the English spelling;

3-      Use more descriptive and precise expressions in the conclusion part;

4-  The title implies that the manuscript is dealing with the Genome-wide identification of strawberry C2H2-ZFPs C1-2i subclass and reveals the potential function of FaZAT10 in abiotic stress. However, the aim of the study is not stated in the abstract and introduction section (Rows 94-98 – PAGE 2).

5-      Inline 216 and 217, change fruit development to fruit  improvement

6-      Arabidopsis thaliana' should be mention the first time, 'A. thaliana' the 2nd time during abstract, introduction ... sections

7-      In, Materials sources and culture conditions section, please mention the sterilization procedure of explants (tissues)

8-      Inline 130, They were mainly distributed in five groups, the sentence is not clear.

9-      Please reference this statement “In cultivated strawberries, the family members of C2H2-ZFPs and their regulatory mechanisms to abiotic stress are largely unknown.

10-  The discussion is not elaborated.

11-  2 g, 5 g, 3 percentage - 4 °C, 2-3 days  ......... Kindly change to two, five, three............

12-  Inline 51 page 2, Kindly change but to In contrast.

Author Response

Dear Editors and Reviewers:

Thank you so much for your letter and for the comments concerning our manuscript entitled “Genome-wide identification of strawberry C2H2-ZFPs C1-2i subclass and reveal the potential function of FaZAT10 in abiotic stress” (ID: ijms-1972817). We appreciate the time and effort that you and the reviewers have dedicated to providing your valuable feedback on our manuscript. Those comments are very valuable and helpful for revising and improving our paper. We have studied all the reviewers’ comments carefully and have made corrections which we hope to meet with approval. Revised portions are highlighted in red using the "Track Changes" function in Microsoft Word in the paper. Below we provide a copy of the reviewers’ comments with our point-to-point responses.

We really appreciate that you give us the opportunity to revise our manuscript. We look forward to hearing a positive response from you.

Best regards,

Dr. Haoru Tang

Professor of Pomology

Sichuan Agricultural University, China

E-mail: htang@sicau.edu.cn

Response to Reviewer 1 Comments

The manuscript entitled "Genome-wide identification of strawberry C2H2-ZFPs C1-2i subclass and reveal the potential function of FaZAT10 inabiotic stress". It is reviewed, this study is comprehensive research. This study is interesting and useful. But there are some deficiencies are in manuscript. By this, minor revision required for article. The paper should be thoroughly revised (typing and grammar mistakes and another mistakes).The manuscript is recommended for publication.

Response: Thank you so much for your comments and suggestions on our manuscript. Your revisions/suggestions have definitely improved the quality of our manuscript. We carefully checked the concerns you pointed, and carefully addressed and revised them.

Minor revisions:

Point 1: Please, explain why did you use this material.

Response 1:

-- Thank you for your attention to this issue.

-- Strawberries have high economic value, Shuangliu County in Sichuan Province is one of the three major strawberry bases in China, octaploid cultivated strawberry is the main cultivar. Most of the Yangtze River flows through the Sichuan Basin. Since July 2022, the Yangtze River basin has suffered the longest drought since records began in 1961. Abiotic stress such as severe drought, salt and low temperatures adversely affect strawberry production, which is the direction of our attention, and octaploid cultivated strawberry is selected as the research object.

Point 2:  Check the English spelling

Response 2:

-- Thank you for pointing this out.

-- We have checked English throughout the paper and revised some words or sentences to make the paper more readable.

Point 3: Use more descriptive and precise expressions in the conclusion part

Response 3:

--Thank you for your comment and suggestion

--As suggested by the reviewer, we have rewritten the conclusion part.

Conclusions

In this study, for the first time, the C2H2-ZFPs in octoploid and diploid strawberry were identified. The C1-2i subclass members which contain two C2H2 domains were further systematically analyzed. These results contribute to our understanding of the evolution of C1-2i subclass in cultivated strawberry. Meanwhile, FaZAT10 of C1-2i subclass was cloned based on stress transcriptome data. Analysis of cis-acting elements in the promoter region and stress induction experiments revealed that FaZAT10 may play an important role in regulating stress signal transduction. Taken together, these results provided new information on C2H2-ZFPs response to abiotic stress in strawberry. In addition, we preliminarily investigated the transcription factor properties of FaZAT10, the specific molecular mechanism of FaZAT10 in abiotic stress needs to be further studied.

Point 4: The title implies that the manuscript is dealing with the Genome-wide identification of strawberry C2H2-ZFPs C1-2i subclass and reveals the potential function of FaZAT10 in abiotic stress. However, the aim of the study is not stated in the abstract and introduction section (Rows 94-98 – PAGE 2).

Response 4:

-- Thank you for the observation and suggestion.

-- As suggested by the reviewer, We have adjusted and revised the last part of the Introduction to make our research purpose more clear.

At present, most of the studies on C2H2-ZFPs in abiotic stress are still focused on model plants such as A. thaliana. In cultivated strawberries, the family members of C2H2-ZFPs and their regulatory mechanisms to abiotic stress are largely unknown. In the present study, the C2H2-ZFPs from cultivated and wild strawberries were identified based on genomic data. Among these C2H2-ZFPs, the phylogenetic relationship, protein physicochemical properties, gene structure, conserved domains, promoter cis-acting elements, and gene collinearity of C1-2i subclass were further analyzed. Then, FaZAT10 from C1-2i subclass was isolated. The expression pattern, subcellular localization, transcriptional activity, as well as the expression partten of FaZAT10 under different abiotic stresses and hormone treatments were further analyzed.  These results improve the characterization of strawberry C2H2-ZFPs family members and provide a research basis for further exploration of how FaZAT10 plays a role in strawberry abiotic regulatory networks.

Point 5: Inline 216 and 217, change fruit development to fruit  improvement

Response 5:

-- Thank you for your comments and suggestions

-- As suggested by the reviewer, We have adjusted and revised to what reviewer mentioned.

In general, the transcript abundance of FaZAT10 was very low in different fruit developmental stages, and it had a higher expression level at little green stage than other fruit developmental stages.

Point 6: Arabidopsis thaliana' should be mention the first time, 'A. thaliana' the 2nd time during abstract, introduction ... sections

Response 6:

-- Thank you for pointing this out.

-- As suggested by the reviewer, We have made the corresponding modification, ‘Arabidopsis thalian’ should be mention the first time, 'A. thaliana' the 2nd time during abstract, introduction and other sections.

Point 7:  In, Materials sources and culture conditions section, please mention the sterilization procedure of explants (tissues)

Response 7:

--Thank you for your comments and suggestions

-- As suggested by the reviewer, We have added the appropriate information.

Octoploid cultivated strawberry 'Benihoppe' (Fragaria × ananassa Duch.), were asexually propagated using runners. The explant surface sterilization procedure was 70% alcohol soaking for 30s, pouring out the alcohol, rinsing with sterile water two times, then soaking with 0.1% HgCl2 for 30 minutes, rinsing with sterile water five times. Finally inoculating the sterilized runners into MS medium (0.5 mg/L 6-BA, 0.1 mg/L IBA) cultured for 45 days and then transplanting to 1/2 MS medium. Plant material were cultured in plant material culture room and greenhouse, under long-day conditions (16 h light, 5,000 lx, 8 h dark, 22°C).

Point 8:   Inline 130, They were mainly distributed in five groups, the sentence is not clear.

Response 8:

-- Thank you for your help and suggestions.

-- As suggested by the reviewer, we added some information to this sentence to make it easier to understand

The 46 C1-2i subclass members from cultivated strawberry and 13 C1-2i subclass members from wild strawberry were mainly distributed in five groups

Point 9-10:  Please reference this statement “In cultivated strawberries, the family members of C2H2-ZFPs and their regulatory mechanisms to abiotic stress are largely unknown;The discussion is not elaborated.

Response 9-10:

--Thank you for your comments and suggestions

--As suggested by the reviewer, We put some details into the discussion to refine the it.

Cultivated strawberry have high economic value and are vulnerable to the negative effects of abiotic stress. Although A. thaliana provides an ideal model for studying abiotic stresses, no C2H2-ZFPs family members have been reported in cultivated or wild strawberries so far

Point 11:  -  2 g, 5 g, 3 percentage - 4 °C, 2-3 days  ......... Kindly change to two, five, three............

Response 11:

-- Thank you for pointing this out.

-- We agree with the reviewer's suggestion and have made corresponding modifications in the paper

Such as: Inline 160, ”7” to “seven”; inline 215, ”6” to “six” etc.

Point 12:  Inline 51 page 2, Kindly change but to In contrast.

Response 12:

-- Thank you for your comments and suggestions

--As suggested by the reviewer, we have rewritten this sentence.

Typically, abiotic stress induces the production of excess reactive oxygen species (ROS) and causes oxidative damage to plants.

Reviewer 2 Report

Manuscript title: Genome-wide identification of strawberry C2H2-ZFPs C1-2i subclass and reveal the potential function of FaZAT10 in abiotic stress

Manuscript ID:  ijms-1972817

Journal:  IJMS

In the current manuscript 126 C2H2-ZFPs from cultivated strawberry were firstly identified using the recently sequenced F. × ananassa genome. Among these C2H2-ZFPs, 46 members containing two zinc finger domains in cultivated strawberry were further identified as C1-2i subclass.

The idea is sound and the manuscript is well written. Abstract is informative. Material and methods contain details which help other researchers to follow easily. Statistical analysis was well performed. Results and discussion are correlated and updated. Academic English language is fine. However, some minor suggestions need to be considered before accepting the current version:

In the abstract, please write the full name of F. × ananassa;

96-98: please delete this sentence;

Kindly write the aim of the study more clear at the end of the introduction section;

Better resolution should be provided for all figures;

Conclusion should be rewritten with no repetition of the abstract.

Author Response

Dear Editors and Reviewers:

Thank you so much for your letter and for the comments concerning our manuscript entitled “Genome-wide identification of strawberry C2H2-ZFPs C1-2i subclass and reveal the potential function of FaZAT10 in abiotic stress” (ID: ijms-1972817). We appreciate the time and effort that you and the reviewers have dedicated to providing your valuable feedback on our manuscript. Those comments are very valuable and helpful for revising and improving our paper. We have studied all the reviewers’ comments carefully and have made corrections which we hope to meet with approval. Revised portions are highlighted in red using the "Track Changes" function in Microsoft Word in the paper. Below we provide a copy of the reviewers’ comments with our point-to-point responses.

We really appreciate that you give us the opportunity to revise our manuscript. We look forward to hearing a positive response from you.

Best regards,

Dr. Haoru Tang

Professor of Pomology

Sichuan Agricultural University, China

E-mail: htang@sicau.edu.cn

Response to Reviewer 2 Comments

In the current manuscript 126 C2H2-ZFPs from cultivated strawberry were firstly identified using the recently sequenced F. × ananassa genome. Among these C2H2-ZFPs, 46 members containing two zinc finger domains in cultivated strawberry were further identified as C1-2i subclass.

The idea is sound and the manuscript is well written. Abstract is informative. Material and methods contain details which help other researchers to follow easily. Statistical analysis was well performed. Results and discussion are correlated and updated. Academic English language is fine. However, some minor suggestions need to be considered before accepting the current version:

Response: Thank you so much for taking your time to review this manuscript. Your comments and suggestions are of great help in improving the quality of our manuscript. We really appreciate all your generous comments and suggestions! According to your advice, we amended the relevant part in manuscript.

Point 1:  In the abstract, please write the full name of F. × ananassa.

Response 1:

-- Thank you for your attention and suggestion

-- As suggested by the reviewer, We have revised the full name of the strawberry, “F. × ananassa” to Fragaria × ananassa

Point 2-3:  96-98: please delete this sentence; Kindly write the aim of the study more clear at the end of the introduction section

Response 2-3:

-- Thank you for your careful review

--As suggested by the reviewer, We rewrote the sentence to make our purpose clearer.

These results improve the characterization of strawberry C2H2-ZFPs family members and provide a research basis for further exploration of how FaZAT10 plays a role in strawberry abiotic regulatory networks.

Point 4:  Better resolution should be provided for all figures

Response 4:

-- Thank you for your comments and suggestions

-- We agree with the reviewer's suggestion and double-checked the figures used in the text, again making sure that the resolution of the figures was 600dpi.

Point 5:  Conclusion should be rewritten with no repetition of the abstract.

Response 5:

-- Thank you for your careful review

-- We agree with this comment. Therefore, We have rewritten the conclusion section.

In this study, for the first time, the C2H2-ZFPs in octoploid and diploid strawberry were identified. The C1-2i subclass members which contain two C2H2 domains were further systematically analyzed. These results contribute to our understanding of the evolution of C1-2i subclass in cultivated strawberry. Meanwhile, FaZAT10 of C1-2i subclass was cloned based on stress transcriptome data. Analysis of cis-acting elements in the promoter region and stress induction experiments revealed that FaZAT10 may play an important role in regulating stress signal transduction. Taken together, these results provided new information on C2H2-ZFPs response to abiotic stress in strawberry. In addition, we preliminarily investigated the transcription factor properties of FaZAT10, the specific molecular mechanism of FaZAT10 in abiotic stress needs to be further studied.